

# Elucidating the diet of the island flying fox (*Pteropus hypomelanus*) in Peninsular Malaysia through Illumina Next-Generation Sequencing

Sheema Abdul Aziz[1,2,3,4], Gopalasamy Reuben Clements[1,2,3,5,6], Lee Yin Peng[5,7], Ahimsa Campos-Arceiz[3], Kim R. McConkey[3,8], Pierre-Michel Forget[2] and Han Ming Gan[5,7]

[1] Rimba, Bandar Baru Bangi, Selangor, Malaysia
[2] UMR MECADEV 7179 CNRS-MNHN, Muséum National d'Histoire Naturelle, Département Adaptations du Vivant, Brunoy, France
[3] School of Environmental and Geographical Sciences, The University of Nottingham Malaysia Campus, Semenyih, Selangor, Malaysia
[4] Centre for Biological Sciences, Faculty of Natural and Environmental Sciences, University of Southampton, Southampton, United Kingdom
[5] School of Science, Monash University Malaysia, Petaling Jaya, Selangor, Malaysia
[6] Kenyir Research Institute, Universiti Malaysia Terengganu, Kuala Terengganu, Malaysia
[7] Genomics Facility, Tropical Medicine and Biology Platform, Monash University Malaysia, Petaling Jaya, Selangor, Malaysia
[8] School of Natural Sciences and Engineering, National Institute of Advanced Studies, Indian Institute of Science Campus, Bangalore, India

Corresponding author
Sheema Abdul Aziz,
sheema@rimbaresearch.org

## ABSTRACT

There is an urgent need to identify and understand the ecosystem services of pollination and seed dispersal provided by threatened mammals such as flying foxes. The first step towards this is to obtain comprehensive data on their diet. However, the volant and nocturnal nature of bats presents a particularly challenging situation, and conventional microhistological approaches to studying their diet can be laborious and time-consuming, and provide incomplete information. We used Illumina Next-Generation Sequencing (NGS) as a novel, non-invasive method for analysing the diet of the island flying fox (*Pteropus hypomelanus*) on Tioman Island, Peninsular Malaysia. Through DNA metabarcoding of plants in flying fox droppings, using primers targeting the *rbcL* gene, we identified at least 29 Operationally Taxonomic Units (OTUs) comprising the diet of this giant pteropodid. OTU sequences matched at least four genera and 14 plant families from online reference databases based on a conservative Least Common Ancestor approach, and eight species from our site-specific plant reference collection. NGS was just as successful as conventional microhistological analysis in detecting plant taxa from droppings, but also uncovered six additional plant taxa. The island flying fox's diet appeared to be dominated by figs (*Ficus* sp.), which was the most abundant plant taxon detected in the droppings every single month. Our study has shown that NGS can add value to the conventional microhistological approach in identifying food plant species from flying fox droppings. At this point in time, more accurate genus- and species-level identification of OTUs not only requires support from databases with more representative sequences of relevant plant DNA, but probably necessitates *in situ* collection of plant specimens to create a reference collection. Although this method

cannot be used to quantify true abundance or proportion of plant species, nor plant parts consumed, it ultimately provides a very important first step towards identifying plant taxa and spatio-temporal patterns in flying fox diets.

# INTRODUCTION

Understanding the contribution of animals to the functioning of rainforests has become an important issue in conservation biology. Conservation studies are now recognizing the need to collect qualitative and quantitative information on trophic relationships between animals and plants, not only to identify potential ecosystem service providers (*Pompanon et al., 2012*; *Hibert et al., 2013*), but also to inform management interventions for threatened species (*Valentini et al., 2009*; *Ando et al., 2013*).

Bats (Order: Chiroptera) provide important ecosystem services such as insect pest suppression, pollination, and seed dispersal (*Fujita & Tuttle, 1991*; *Kunz et al., 2011*). Characterising their diet is a fundamental step towards understanding their ecological roles. In the Old World, fruit bats such as flying foxes (Pteropodidae: *Pteropus* spp., *Acerodon* spp.; *Kingston, 2010*) have become increasingly threatened by hunting for bushmeat and medicine (*Mildenstein, Tanshi & Racey, 2016*). Identifying their diet and roles as ecosystem service providers can help strengthen arguments for their protection. It will also help us understand the wider implications of large-scale flying fox extinctions, as these giant bats are known to interact with plants on a large landscape scale, performing ecological roles over vast transboundary areas (*Epstein et al., 2009*). Flying foxes are likely to be particularly important players in island ecosystems where they often serve as keystone pollinators and seed dispersers both within and between islands (*Cox et al., 1991*; *Banack, 1998*; *McConkey & Drake, 2007*; *McConkey & Drake, 2015*), and where maintaining their numbers at high densities is necessary for the survival of plant communities (*McConkey & Drake, 2006*). Such data are also important to understand the drivers and potential mitigation strategies for conflicts between fruit bats and humans (*Aziz et al., 2016*; *Aziz et al., in press*).

Whilst in-depth, comprehensive dietary/foraging studies have been conducted for certain flying fox species, particularly in Australia (e.g., *Boulter et al., 2005*; *Williams et al., 2006*), Oceania (e.g., *McConkey & Drake, 2006*; *Luskin, 2010*), Japan (e.g., *Nakamoto, Kinjo & Izawa, 2007*; *Nakamoto, Kinjo & Izawa, 2009*; *Lee et al., 2009*), South Asia (e.g., *Mahmood-ul-Hassan et al., 2010*; *Sudhakaran & Doss, 2012*), and Indian Ocean islands (e.g., *Nyhagen et al., 2005*; *Oleksy, Racey & Jones, 2015*), the diets of Southeast Asian species, which are some of the most threatened due to the additional threat of commercial hunting (*Mildenstein, Tanshi & Racey, 2016*), remain largely unknown. Indeed, apart from a few studies in the Philippines (*Reiter & Curio, 2001*; *Mildenstein et al., 2005*; *Stier & Mildenstein, 2005*), Thailand (*Weber et al., 2015*), and Myanmar (*Win & Mya, 2015*), all other dietary and foraging studies on Southeast Asian Pteropodidae have focused on the smaller
pteropodids (e.g., *Hodgkison et al., 2004*; *Fletcher, Zubaid & Kunz, 2012*; *Bumrungsri et al., 2013*; *Stewart, Makowsky & Dudash, 2014*). This is of particular concern given that out of the 67 flying fox species listed on the IUCN Red List, almost half (30 species i.e., 45%) are found in Southeast Asia (*IUCN, 2016*).

Due to the nocturnal and volant nature of bats, invasive analyses (by capturing individuals) or indirect methods (by collecting droppings) have traditionally been used to study their diets. Microscope analyses of pteropodid faeces have provided insights into their diet (*Bumrungsri, Leelapaibul & Racey, 2007*), as well as their roles in pollination (*Bumrungsri et al., 2013*) and seed dispersal (*Sritongchuay et al., 2014*). However, all these studies have relied on physical identification of food plant species—either through direct observations of foraging bats, or microhistological identification of seeds, pollen, fruit fibres and leaf fragments in faeces and ejecta (chewed-up pellets of plant parts spat out by bats after swallowing the juice or soft pulp; *Nyhagen et al., 2005*; *Long & Racey, 2007*). The successful use of these approaches relies on several important factors such as accessibility and visibility of foraging bats for the former method, and also the availability of expert botanical knowledge or resources such as reference collections. Another limitation of these conventional approaches is that they require physically identifiable remains to be expelled by the bats; any plant parts that were consumed or expelled solely in liquid form will be missed out in the analysis (*Pompanon et al., 2012*). Studies on the foraging ecology of wide-ranging species such as flying foxes also require the use of expensive, hi-tech equipment such as GPS collars, which is often not feasible.

Although molecular analysis of pteropodid diets can potentially be used to overcome the obstacles outlined above, this approach has yet to be applied. DNA analyses of faeces collected non-invasively have already been conducted to determine the herbivorous diets of animals such as primates (*Bradley et al., 2007*; *Quéméré et al., 2013*; *Srivathsan et al., 2016*), marmots, bears, capercaillies, grasshoppers, molluscs, slugs (*Valentini et al., 2009a*), pigeons (*Ando et al., 2013*) and tapirs (*Hibert et al., 2013*), but this has never before been attempted for pteropodids in the Palaeotropics. To date, molecular analyses of bat diets in the Old World have only been conducted for insectivorous species (e.g., *Clare et al., 2009*; *Razgour et al., 2011*; *Zeale et al., 2011*). In fact, to our knowledge, the only successful attempt to identify the diet of plant-visiting bats through molecular analysis has been done by one study in the Neotropics (*Hayward, 2013*).

We evaluated the utility of Illumina Next Generation Sequencing (NGS) to identify plant species present in the droppings of the island flying fox (*Pteropus hypomelanus*) from Tioman Island in Peninsular Malaysia. In addition, we evaluated the potential of NGS analysis in complementing or even replacing conventional microhistological analysis to elucidate flying fox diets. First, we created a site-specific reference collection of potential flying fox food plants—DNA sequencing of these plants facilitated the construction of a phylogenetic tree to support morphology-based identification of these food plant species and primer design for NGS. Next, we assessed the feasibility of extracting plant DNA from flying fox droppings, and evaluated whether DNA sequences obtained from NGS could be matched with those from online and site-specific DNA reference databases. Finally,

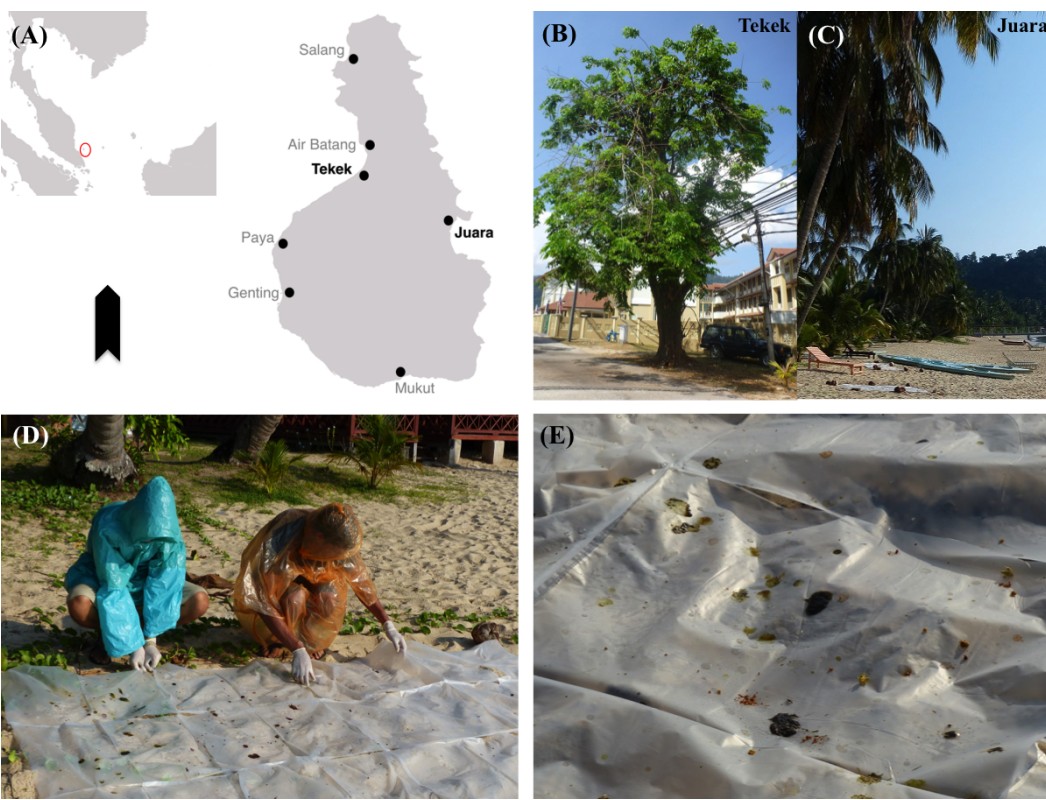

**Figure 1** **Map of study area and images of sampling site and method.** (A) Map of Tioman showing sampling sites Tekek & Juara. (B, C) Examples of flying fox roosts sampled in Tekek & Juara. (D) Collecting droppings from roosts. (E) Close-up of droppings.

we compared the performance of NGS with conventional microhistological analyses to identify food plant species from flying fox droppings.

## MATERIALS AND METHODS

### Study site

We conducted this study on Tioman Island (2°48′38″N, 104°10′38″E; 136 km$^2$; Fig. 1A), located 32 km off the east coast of Peninsular Malaysia in the State of Pahang. This research was approved by the Economic Planning Unit of Malaysia (Permit number: 3242). Much of the island inland is still covered by primary tropical rainforest, which has been designated as Pulau Tioman Wildlife Reserve (83 km$^2$). It has a hilly topography, with flat areas only along the coast (*Abdul, 1999*). The area designated as a wildlife reserve is composed of lowland mixed dipterocarp forest and hill dipterocarp forest. Most forested areas are still inaccessible due to the rugged topography, with many steep slopes and rocky outcrops (*Latiff et al., 1999*). The climate is tropical, uniformly warm and humid throughout the year (*Hasan Basyri et al., 2001*), but the island experiences the northeast monsoon from November to March (*Bullock & Medway, 1966*).

There are currently seven villages on the island, situated along the coastline (Fig. 1A). The majority of the local people are Muslim, and therefore due to religious dietary restrictions

do not hunt bats for food or medicine (*Aziz, 2016*). As the island's marine area is also a designated Marine Park and a popular tourist destination, many of the local people are heavily involved in the tourism industry (*Abdul, 1999*).

## Study species

The island flying fox (*Pteropus hypomelanus*), also known as the variable flying fox and the small flying fox, roosts gregariously, forming colonies of up to 5,000 individuals. It is a widespread insular species, considered to be abundant throughout a distribution range that extends from the Maldives and Indian islands in the west to Melanesia in the east. Because of this, it is considered to be Least Concern on a global scale by the IUCN Red List; however, populations are decreasing (*Francis et al., 2008*; *Olival, 2008*), and the species is now listed as endangered on the Malaysian Red list (*DWNP , 2010*).

On Tioman, the island flying fox can be found roosting permanently in two villages: Tekek, on the west coast, and Juara, on the east coast (Fig. 1A), and forages throughout the island (*Medway, 1966*; *Ong, 2000*). Monthly roost counts conducted during March–October 2015 using a thermalscope (Pulsar Quantum HD38S) yielded estimated ranges of 2,178–5,385 individuals for the entire island (see Fig. S1 for further details). Local people have reported that the flying foxes do forage in other villages on the island.

## Flying fox dropping collection

Collection of flying fox droppings took place once a month during March-October 2016 (i.e., eight months). Samples of droppings consisting of faeces and ejecta were collected for three mornings in the last week of each month from three separate day roosts in Juara (east coast) and two separate day roosts in Tekek (west coast). The number of roosts and sampling days were determined based on species accumulation curves of pollen morphospecies that were detected through preliminary microhistological analysis in June 2014. Program EstimateS (version 9.1.0; http://viceroy.eeb.uconn.edu/estimates/) indicated that sampling completeness (i.e., observed/estimated number of species; *Soberón, Llorente & Onate, 2000*) was around 97% using this sampling regime.

In Juara, three suitable roost trees (Fig. 1B) for sampling were selected based on accessibility and also on the highest/largest amount of faecal/ejecta splatter produced under the roost, in order to maximise sample yield. As flying foxes often shifted roosts or temporarily abandoned degraded roosts, this meant that sometimes different roosts were sampled in each location every month or even every morning, although most roosts were consistently sampled each month due to their constant high occupancy and best accessibility.

Plastic sheets measuring $0.8 \times 1.0$ m were placed under each roost after dark, once the bats had exited the roost to forage. The roosts were then visited the next morning for collection starting at 0700 h and ending at 1200 h (bats typically returned to the roosts around 0500–0600h); the plastic sheets were carefully moved away from the roost to a clear area for processing (Fig. 1C). As it was often difficult to differentiate faeces from ejecta, both were collected and analysed equally as 'droppings' (Fig. 1D). Droppings collected for processing were selected based on unique colour and texture, as this was assumed

to be representative of plant diversity in the bats' diet. Following the approach used by *Stier & Mildenstein (2005)* based on short gut-passage time for flying foxes (12–34 min; *Tedman & Hall, 1985*), we assumed that each bat voided its last meal once, and therefore each dropping represented a different individual's food choice. We devised our own novel collection protocol where droppings were collected by swabbing them with a cotton bud, then placing each individual dropping into a 5 ml Eppendorf tube containing ∼1,000 μl of 95% ethanol. These tubes were then kept cool in the field, either by storing in a conventional freezer or by using a portable cooler box with ice packs, for 1–3 days before being transported off the island and then stored in a −80 °C freezer.

In order to simultaneously test the utility of NGS and compare it with conventional approaches, we collected two duplicate sets of 10 individual droppings from one single roost in Juara village during a single morning on 6 May 2015. One sample set was then kept in a conventional fridge for microscope analysis, whilst the other set was stored in the −80 °C freezer for molecular analysis.

## Site-specific plant reference collection

We first checked a published list of genera of known food plants for *Pteropus* across its range (*Marshall, 1985*), cross-checked this against a preliminary checklist of seed plants for Tioman (*Latiff et al., 1999*), and also obtained information on possible flying fox food plants through talking to local people in Juara. We then searched for genera of similar plants in and around the two villages with the aid of a local plant expert. The botanical identification of plants (at least to genus) were subsequently verified by a trained botanist familiar with plants from the region. When we found an individual plant from one of these genera, we recorded its GPS location and collected pollen, fruit, and/or seeds if it was flowering or fruiting. We also collected 3–5 mature leaves for DNA extraction. The leaves were stored in Ziploc bags with silica gel under cool conditions to retard decomposition rates. Leaf samples from 19 different plant species were obtained for this purpose, constituting a preliminary library (Table S1).

Genomic DNA was extracted from approximately 25 mg of one leaf from each plant species using DNAeasy Plant Mini Kit (Qiagen, Halden, Germany) according to the manufacturer's protocols. Primers targeted the *rbcL* gene, a protein-coding gene associated with the chloroplast genome of all living plants. DNA amplifications were performed in a mastermix containing 1 μL of DNA, 25 μL of OneTaq Quick-Load 2X Master Mix with Standard Buffer, (New England Biolab, Ipswich, MA, USA), 1 μL of 10mM forward primer rbcLaf-M13, 1 μL of 10 mM reverse primer rbcLa-revM13 (Table S2), and 22 μL of nuclease-free water. The PCR protocol was started with an initial denaturation step for 30 s at 94 °C, followed by 30 cycles of 30 s at 94 °C, 30 s at 48 °C, 40 s at 68 °C, and final elongation for 2 min at 68 °C. The PCR products were purified using 0.8× volume ratio of Agencourt Ampure XP beads (Beckman Coulter, Inc). The purified samples were sent to 1st BASE laboratories (http://www.base-asia.com/) for Sanger sequencing. The sequencing results were quality trimmed using CodonCode TraceViewer (http://www.codoncode.com/TraceViewer/) and aligned using MAFFT version 7.0 (*Katoh & Standley, 2013*).

A phylogenetic tree (Fig. S4) was constructed to support morphology-based identification of site-specific reference plants. Additional *rbcL* gene sequences that displayed high similarity to the sequenced plant specimens were mined from the BOLD/NCBI databases, aligned with MAFFT version 7.0 and trimmed using TrimAl version 1.9 optimized for maximum likelihood tree construction (*Capella-Gutiérrez, Silla-Martínez & Gabaldón, 2009*). The trimmed alignment was subsequently used to construct a maximum likelihood tree with FastTree using the –nt (nucleotide) and –gtr (generalised time-reversible model) setting.

## Laboratory procedure

Based on the alignment of *rbcL* gene fragments from the site-specific reference plant DNA (Fig. S2), we designed a primer pair targeting 198 bp of the *rbcL* gene (minus primer sequence) using Primer3 (http://bioinfo.ut.ee/primer3-0.4.0/) on default settings (Table S2). Prior to primer synthesis, partial Illumina adapter sequences were added to the 5′ end of the designed primers, rbcL-357F and rbcL-556R, to allow barcoding and sequencing on the Illumina platform.

Individual droppings were pooled according to roost ($n = 5$, 2 in Tekek and 3 in Juara) and month ($n = 8$), creating 40 separate mixtures for analysis. The tubes containing the daily samples were first vortexed for 2 min to homogenise the content and subsequently, 1,000 µL of the sample was pipetted into another tube to form the mixture. Next, 100 µL of the mixture was used for DNA extraction similarly using DNAeasy Plant Mini Kit (Qiagen, Halden, Germany) instead of a stool-specific DNA extraction kit to improve the recovery of plant-derived DNA from faecal samples.

PCR reaction was performed using IlluM_rbcLF and IlluM_rbcLR. The 20 µL PCR cocktail consists of 10 µL *Q5* Hot Start High-Fidelity 2× *Master Mix* (New England Biolab, Ipswich, MA, USA), 1 µL each of 10 µM forward and reverse primer, 1 µL gDNA and 7 µL nuclease-free water. All reactions were performed in a Veriti® 96-Well Fast Thermal Cycler with the following protocol: initial denaturation for 30 s at 98 °C, 25 cycles of 10 s at 98 °C, 30 s at 55 °C and 10 s at 65 °C, with a final 1 min extension at 65 °C. The PCR product was purified using 0.8× vol. ratio Agencourt Ampure XP beads (Beckman Coulter, Inc). Then, 1 uL of Index 1 and Index 2 primers from Nextera XT kit were added to 3 uL of purified PCR product and combined with 5 uL of *Q5* Hot Start High-Fidelity 2X *M aster Mix* (New England Biolabs, Ipswich, MA). The PCR protocol was as followed: initial denaturation for 30 s at 98 °C, 8 cycles of 10 s at 98 °C and 1 min at 65 °C, with a final 1 min extension at 65 °C.

## Sequence analysis and taxon assignation

The purified amplicons containing the full length Illumina adapter and appropriate unique barcode were then quantified using KAPA Library Quantification kit (Kapa Biosystems, CapeTown, South Africa) on the EcoRealTime PCR system (Illumina, San Diego, CA, USA). Based on the qPCR data, the amplicons were normalised, pooled and subsequently sequenced on the MiSeq (2 × 250 bp paired-end run) located at the Monash University Malaysia Genomics Facility.

Illumina Nextera adapters and primer sequences of the reads were trimmed off using Trimmomatic v 0.33 and FastX trimmer, respectively (*Bolger, Lohse & Usadel, 2014*; http://hannonlab.cshl.edu/fastx_toolkit/). The trimmed paired-end reads were then merged using PEAR (*Zhang et al., 2014*) using default settings. Dereplication, singleton removal and Operationally Taxonomic Unit (OTU) clustering (default setting of 97% identity clustering; -id 0.97) were performed using the pipeline implemented in UPARSE (*Edgar, 2013*). The filtered OTUs were translated into protein sequence and manually inspected for translated sequence containing stop codon(s), which were removed from subsequent analyses. Taxonomic assignment of OTUs was conducted by matching sequences against those from the BOLD/NCBI non-redundant nucleotide database as of December 2016. The BLASTn output for each OTU was downloaded in "XML" format and imported into MEGAN6 (*Huson et al., 2016*) to calculate their Lowest Common Ancestor (LCA). LCA predicts the taxonomic rank of OTUs based on matches to a set of reference lineages such that the predicted taxonomic rank is shared by all matches within user pre-defined parameters. The implemented parameters for MEGAN6 were: Minimum Blast bit score = 200, Max expected $E$-value = 0.01, Min Percent Identity = 90.0, Top percent (the percentage of the best score a match needs to lie within) = 10 and Weighted LCA% = 75. To estimate the OTU relative abundance in each sample, reads were mapped to the filtered OTUs via USEARCH (97% similarity cut-off) and normalised to 10,000 reads (*Caporaso et al., 2010*).

## Microhistological analysis

For the 10 dropping samples collected in May, we sent one set for NGS analysis (following the protocol above) and used another set for microscope analysis. For the latter, we first manually broke up the dropping contents in the tube to produce a relatively more representative liquid sample. We then droppped 1–3 drops of this liquid onto a microscope slide using a pipette. Fuchsin jelly was added to this in order to stain pollen grains within the dropping, a slip cover was placed on top, and the jelly was then melted over an open flame, sealing the slip cover to the slide. The slide was then cooled down in a conventional fridge in order to allow the jelly to solidify again before examination.

Once cooled, we placed the slide under a conventional light microscope (Leica DM E) and first examined it using 10/0.25 magnification in order to detect pollen grains and other plant parts. We used a self-made reference collection as well as photos from *Start (1974)*, S. Bumrungsri (http://www.seabcru.org/seabcru-resources) and *Mohamed (2014)* to identify pollen and other plant parts. When necessary, we used higher magnification (40/0.65) to view the pollen grains. We used just 'presence/absence' to assess pollen to avoid quantification biases towards species that naturally produce greater amounts of pollen. Following *Thomas (2009)*, we considered a species present if we found three or more pollen grains on a single slide.

## RESULTS

### Site-specific plant reference database

For the 19 specimens in the site-specific plant reference database, the phylogenetic tree (Fig. S4) based on a 689 bp aligned *rbcL* gene region recovered the monophyly of several plant genera with high SH-like local support (>0.90). However, the sister grouping among genera in general was not well-supported, possibly due to the lack of comprehensive plant taxon sampling, and the use of a single genetic marker. The *rbcL*-based phylogeny generally supported the morphology-based identification of reference specimens of potential flying fox food plants, with 15 specimens correctly identified at least to genus level as shown by their monophyletic clustering with plant species of the same genus (Table 1). Sample PTMN02 (identified as *Streblus asper* based on morphology) was an interesting exception as it formed a sister group with the *Ficus* clade (SH-like local support = 0.938), but not with other congeners. *rbcL* sequences representing the plant species collected on Tioman were scarce in the BOLD/NCBI databases, suggesting these databases have insufficient sequence representation that can affect taxonomic assignment of plant DNA in the droppings.

### Feasibility of using NGS to study flying fox diet

With our newly designed *rbcL* primer, we were able to successfully extract, amplify, and subsequently identify plant DNA from all of the collected flying fox droppings. Initially, a total of 160 Operationally Taxonomic Units (OTUs) were recovered from the sequencing reads, of which 29 OTUs (Table 1) were retained after filtering based on cumulative relative abundance (>0.5%) and presence of stop codon(s) in reading frame. Using a conservative LCA approach, we identified at least three different plant genera and at least 14 plant families from the droppings (Table 1). In addition, 8 OTUs matched with specimens from the site-specific plant reference collection.

Based on sampling completeness (calculated using EstimateS) for OTU relative abundance data from five roosts (data pooled over three days) per month using Chao 1 species richness estimator (good for datasets skewed towards low abundance classes; *Chao, 1984*), sampling completeness was relatively high for the months March (99%), April (100%), June (100%), August (100%), September (88%), and October (96%). However, sampling completeness could be improved for May (55%) and July (79%). The month of May, which had the lowest overall sampling completeness, also had the highest number of droppings collected (Table S3).

The results from our NGS analysis of island flying fox droppings over eight months suggest that the diet at both Juara and Tekek during this time was dominated by four different plant taxa (Table 1; Fig. 2): OTU 1 (*Ficus* sp.), OTU 3 (likely to be *Mangifera indica* based on sequence match with site-specific plant reference collection) and OTUs 4 & 5 (Rubiaceae). Spatio-temporal patterns in the relative abundance of these four taxa in the diet were observed during the sampling period (Fig. 2). For example, OTU 5 appeared to be consumed in similar proportions at both Juara and Tekek across all months whereas OTU 4 was consistently consumed in low proportions in Tekek yet consumed irregularly in Juara over the same period (Fig. 3). Even between different roosts in the same site, spatio-temporal differences were observed, such as for OTU 7 (Fig. S5).

**Table 1** Summary information of 29 OTUs detected in flying fox droppings over eight months (Mar–Oct 2015) in Tioman Island, Malaysia: OTU identities (genus in bold) based on matches with plant sequences from online reference database using the Least Common Ancestor approach, identity of plant family, OTU relative abundance (Sum of reads for particular OTU/sum of reads for ALL OTUs * 100%) and matches with sequences from plant reference specimens.

| OTU No. | Lowest common ancestor (LCA)[a]: order/family/sub-family/genus | Family | Relative abundance | Plant reference specimen match at 100% identity (code; Table 1) |
|---|---|---|---|---|
| OTU 1 | *Ficus* | Moraceae | 66.3 | *Ficus* sp. (PTMN11/PTMN22) |
| OTU 3 | Anacardiaceae | Anacardiaceae | 15.43 | *Mangifera indica* (PTMN20) |
| OTU 4 | Ixoroideae | Rubiaceae | 5.33 | |
| OTU 5 | Naucleeae | Rubiaceae | 2.64 | |
| OTU 8 | Lamiales | ? | 2.01 | |
| OTU 6 | *Diospyros* | Ebenaceae | 1.67 | |
| OTU 7 | Moraceae | Moraceae | 1.4 | |
| OTU 57 | Anacardiaceae | Anacardiaceae | 0.64 | |
| OTU 9 | Muntingiaceae | Muntingiaceae | 0.6 | |
| OTU 110 | *Ficus* | Moraceae | 0.57 | |
| OTU 11 | Myrtoideae | Myrtaceae | 0.42 | *Syzygium* sp. (PTMN08/PTMN09/PTMN17) |
| OTU 12 | Arecoideae | Arecaceae | 0.42 | *Cocos nucifera* (PTMN07) |
| OTU 13 | *Terminalia* | Combretaceae | 0.4 | *Terminalia catappa* (PTMN19) |
| OTU 15 | Malpighiales | ? | 0.37 | |
| OTU 103 | *Ficus* | Moraceae | 0.25 | |
| OTU 10 | Moraceae | Moraceae | 0.23 | |
| OTU 17 | Malvaceae | Malvaceae | 0.21 | *Durio zibethinus* (PTMN16) |
| OTU 120 | *Ficus* | Moraceae | 0.16 | |
| OTU 21 | Salicaceae | Salicaceae | 0.13 | |
| OTU 16 | Rutaceae | Rutaceae | 0.12 | |
| OTU 18 | Fabids | ? | 0.12 | |
| OTU 22 | Chrysobalanaceae | Chrysobalanaceae | 0.11 | |
| OTU 142 | Ixoroideae | Rubiceae | 0.1 | |
| OTU 89 | Anacardiaceae | Anacardiaceae | 0.09 | |
| OTU 23 | Annonaceae | Annonaceae | 0.07 | |
| OTU 19 | Pentapetalae | ? | 0.06 | *Strombosia* sp. (PTMN06) |
| OTU 126 | Anacardiaceae | Anacardiaceae | 0.05 | |
| OTU 147 | Naucleeae | Rubiaceae | 0.05 | |
| OTU 20 | Lamiaceae | Lamiaceae | 0.05 | *Vitex pinnata* (PTMN05) |

**Notes.**

[a]LCA paramaters: Min score = 200, Max expected = 0.01, Min percent identity = 0.0, Top Percent = 10, Weighted LCA% = 80.

### Performance of NGS vs. microhistological analysis

Microscope analysis identified two plant taxa in flying fox droppings (Table 2). Out of 10 individual droppings, three contained durian (*Durio* sp.) pollen. All the other droppings contained fig parts exclusively; no other plant parts were detected. Durian pollen occurred at extremely low abundance; in all cases, only 3–4 grains were detected per slide. No other

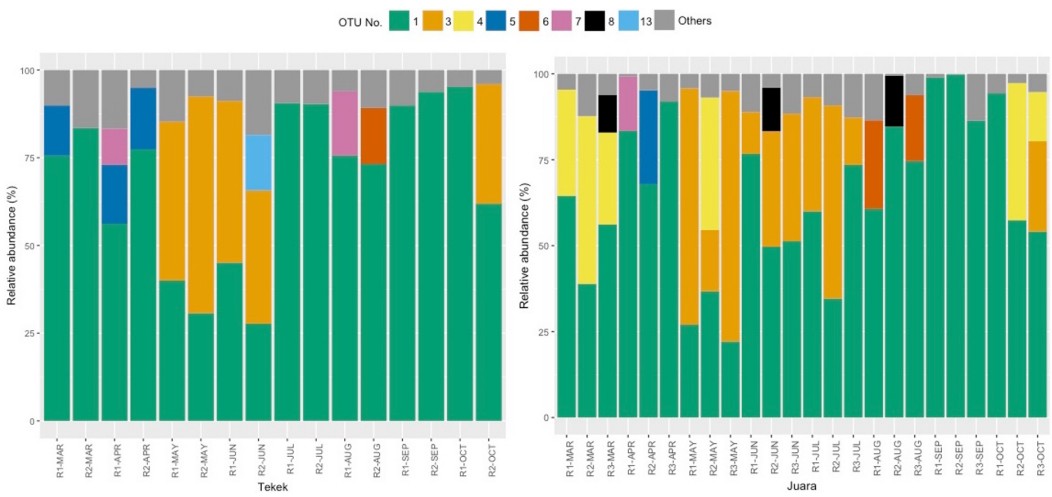

**Figure 2  Relative abundance of 8 OTUs detected in flying fox droppings.** Relative abundance of 8 OTUs detected in flying fox droppings across 8 months (Mar–Oct 2015) at two different roosting sites on Tioman Island, Tekek (two roosts) and Juara (three roosts). OTU 1, Ficus; OTU 3, Anacardiaceae; OTU 4, Rubiaceae; OTU 5, Rubiaceae; OTU 6, Diospyros; OTU 7, Moraceae; OTU 8, Lamiales; OTU 13, Terminalia; Others, pooled OTUs with <5% relative abundance at each roost.

**Table 2  Comparative effectiveness of microhistological vs. NGS analyses in identifying plants from two duplicate sets of 10 samples of flying fox droppings collected on 6 May 2015.** Plant ID (probable genus/family) for the microhistological analysis was based on visual identification from our plant reference collection, while plant ID for the NGS analyses was based on NGS sequence matches with online reference plant databases and DNA extracted from our plant reference collection.

| Plant ID | Microhistological analysis | | | | | | | | | | NGS analysis | | | | | | | | | |
|---|---|---|---|---|---|---|---|---|---|---|---|---|---|---|---|---|---|---|---|---|
| | 1 | 2 | 3 | 4 | 5 | 6 | 7 | 8 | 9 | 10 | 1 | 2 | 3 | 4 | 5 | 6 | 7 | 8 | 9 | 10 |
| 17 (*Durio*) | x | | | | | | x | | x | | | | | | x | | | x | x | |
| 1 (*Ficus*) | | x | x | x | x | x | x | x | x | x | x | x | x | x | x | x | x | x | x | x |
| 3 (*Mangifera*) | | | | | | | | | | | x | x | x | x | x | x | x | x | x | x |
| 19 (*Strombosia*) | | | | | | | | | | | | x | x | x | | | | | x | |
| 13 (*Terminalia*) | | | | | | | | | | | x | x | | | x | x | x | | | |
| 12 (Arecaceae) | | | | | | | | | | | x | x | x | x | x | x | x | x | x | x |
| 5 (Rubiaceae) | | | | | | | | | | | | x | | | x | | | | | |

pollen or plant parts were detected. NGS identified the same two plant taxa detected by microhistological analysis, and further identified an additional six plant taxa. However, *Durio* was not detected in the same samples as those identified via microscope.

## DISCUSSION

Our study is the first to describe the diet of the island flying fox, which was previously unknown. To our knowledge, this is also the first use of NGS to identify plant taxa in the diet of a pteropodid, which has been difficult to characterise due to these animals' volant nature, large home ranges and nocturnal foraging behaviour. Figs consistently formed the highest amount of plant taxa detected in the droppings each month, at both sampling sites. This strongly suggests that figs compose the core diet of flying foxes on the island. It is thus

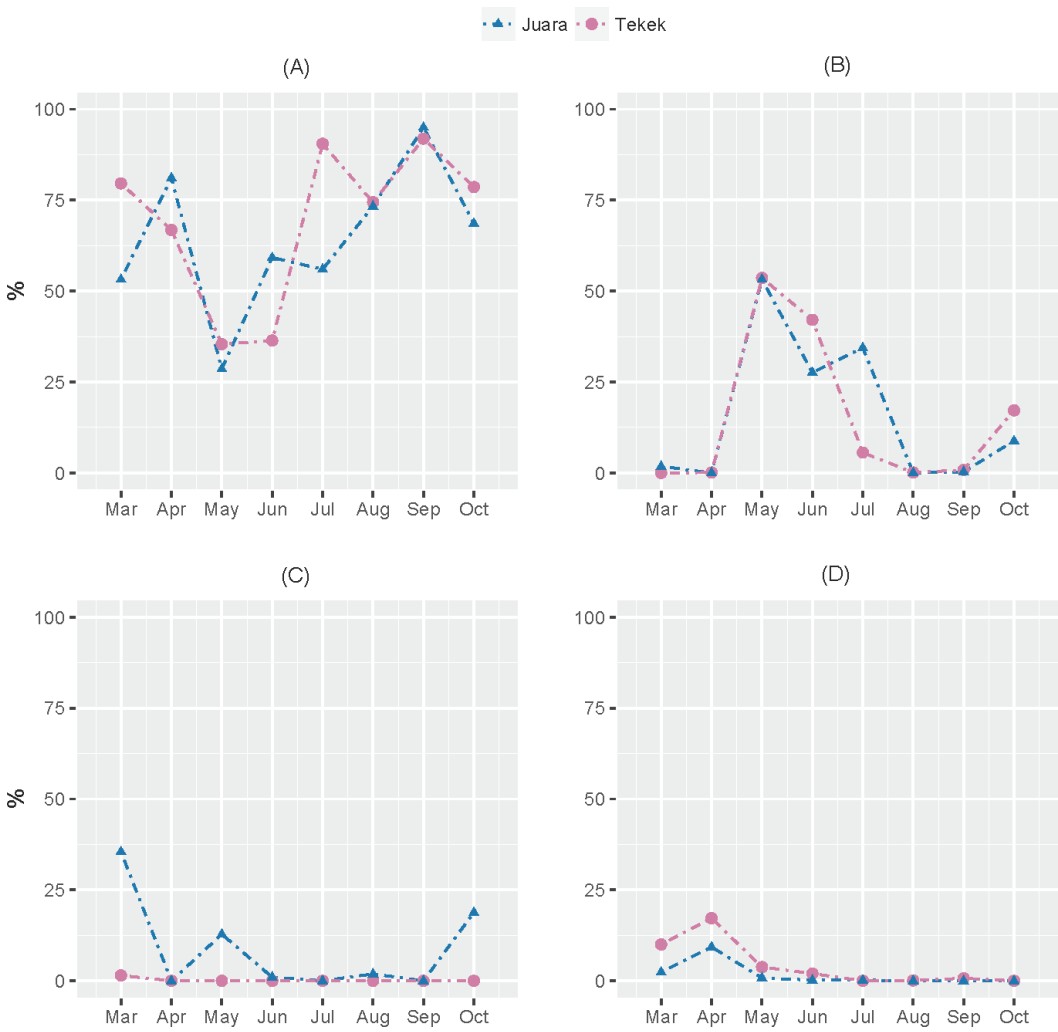

**Figure 3** **Spatio-temporal trends in predicted consumption of the top four most dominant plant taxa.** Spatio-temporal trends in predicted consumption of the top four most dominant plant taxa detected in flying fox droppings during March–October 2015 on Tioman Island based on NGS analysis. (A) OTU 1, *Ficus*; (B) OTU 3, Anacardiaceae; (C) OTU 4, Rubiaceae; (D) OTU 5, Rubiaceae.

highly likely that the island flying fox plays a key role in dispersing fig seeds throughout Tioman, making these bats important keystone species for the island (*Cox et al., 1991*; *McConkey & Drake, 2015*); future studies on seed dispersal and germination are required to confirm this.

## NGS is a reliable tool to study flying fox diet

We have demonstrated that identification of plant taxa to family level is generally possible based on the partial sequence of *rbcL* using the LCA approach. In addition, some OTUs in our study were successfully assigned to the genus level. In order to be conservative, however, we avoided assigning most OTUs to species level, unless there were matches with BOLD/NCBI database sequences and site-specific reference plant sequences. As

species-level plant identification based solely on NGS is not straightforward, the use of a site-specific food plant DNA reference database is vital to help identify plant species in flying fox diets.

Other genes have been successfully used to identify plant species in animal diets. For example, *Valentini et al. (2009a)* found the *trnL* intronic region to be effective for Asian mammals, birds, and invertebrates, identifying 50% of the plant taxa found in the diets of these animals to species level. The same approach has been used for European bison (*Kowalczyk et al., 2011*), alpine chamois (*Raye' et al., 2011*), and red-headed wood pigeons (*Ando et al., 2013*). The *trnL* intron has been previously reported to evolve more than three times faster than the protein-coding *rbcL* gene, thus potentially harbouring more variation and phylogenetic signal per base pair (*Gielly & Taberlet, 1994*). However, we chose the *rbcL* gene instead of the P6 loop of the *trnL* intron to study flying fox diet, as the *rbcL* gene is currently one of the two genes (the other being *matK*) that is increasingly being used for plant species identification. This is useful for studying plants originating from a less-studied region of high biodiversity such as Southeast Asia. Unlike the *trnL* database, the *rbcL* database is consistently growing in the BOLD database, and as a result is also likely to be represented by more sequences from properly identified and vouchered specimens. In addition, given that *rbcL* is a protein-coding gene, it enables the screening of erroneous OTU resulting from sequencing and/or amplification errors based on the presence of stop codon(s) in the translated reading frame. Instead of using recently developed and robustly tested primers (e.g., *Little, 2014*), we designed a new set of *rbcL* minibarcode primers due to the current lack of *rbcL* sequence representation for plant species from Malaysia or specifically, Tioman. One caveat of this approach would be the possibility of preferential primers binding to known diet items instead of unexpected ones. However, our results seem to indicate that our newly designed primers were, to a certain extent, capable of recovering OTUs belonging to a wide variety of plant families, in part due to the diverse representation of plant reference specimens that contributed to the expanded taxon coverage of our primers. Future studies aiming to achieve greater power of identification of plants in flying fox diets could consider using more than one target region (e.g., *Hibert et al., 2013*; *Clare, 2014*) coupled with an additional *in silico* PCR optimisation step using ecoprimers to improve the reliability and universality of the newly designed primers (*Riaz et al., 2011*). Also, in order to completely eliminate bias associated with PCR, metagenomic shotgun sequencing could be performed, albeit at a relatively higher cost depending on the required sequencing coverage. (e.g., *Srivathsan et al., 2015*; *Srivathsan et al., 2016*).

## NGS can complement microhistological analysis

Our results showed that NGS can provide greater insights into the diet of flying foxes than conventional microhistological approaches by detecting a wider range of plant taxa, thus highlighting the utility and discriminatory potential of the newly designed *rbcL* primers to study flying fox diets. More importantly, the use of NGS allowed us to identify plant species even when no physical plant parts were found in the flying fox droppings. The plant genera and families detected from NGS have also been recorded by botanists as being present on

Tioman, including the top four genera/families detected most abundantly in the droppings (*Latiff et al., 1999*; *Mohd. Norfaizal et al., 2014*).

In our study, attempts to use microscope analysis to identify plant parts in droppings proved to be challenging, as no pre-existing reference collection was available. Building our own microhistological reference collection for Tioman was time-consuming and labour-intensive—and the resulting collection often did not match up with the plant parts found in the flying fox droppings. However, obtaining DNA from plant specimens is still necessary to narrow down the identity of OTUs to species level. Indeed, 8 out of 29 OTUs had 100% matches to the sequences of plant specimens collected from the study site, highlighting the importance of building a comprehensive local sequence library beforehand, preferably specific to one's particular study site.

It is important to note that NGS did not detect *Durio* in the same individual droppings as those identified via microscope. This is likely due to the low abundance of this plant taxon in the droppings affecting detection probability, especially since the NGS analysis used a more general primer that was not specific to *Durio*. This pollen detection probability is another caveat to be aware of; *Scanlon et al. (2014)* have cautioned that faecal subsampling methods can potentially lead to inaccurate detection of pollen in dietary studies, regardless of which method is used.

## Caveats

Our sample collection method in the field, selecting only for droppings with unique colour and texture, may have introduced a bias that could result in underestimating the relative abundance of OTUs in the droppings. In particular, sampling completeness for the months May, June and July were relatively low, showing that more roosts and/or days needed to be sampled in order to obtain a complete representation of diet for these months. However, roost count data (Fig. S1) show that this high diversity in diet was not influenced by population abundance. Instead, perhaps diet choice and/or food resource diversity were relatively higher during this period compared to the rest of the year. Future studies should aim to collect every single dropping found underneath a roost to improve representativeness.

Given the potentially short flying fox gut passage times (*Tedman & Hall, 1985*), droppings collected from day roosts in the morning may also bias the analysis results towards food items that were consumed only at the end of the foraging period (*Schmelitschek, French & Parry-Jones, 2009*). Food plants that were consumed during the start or middle of the evening may not have been detected by our methods. Although *Banack & Grant (2002)* observed flying foxes returning to food resources that were foraged upon earlier, before then returning to day roosts, this is still a potential caveat to bear in mind. For example, primates are known to exhibit temporal patterning in diet choice, structuring their diet throughout their foraging period with different food items; it is believed that this is due to how different foods are processed, and give energy, at different rates, and therefore helps to ensure that the animals maintain high energy levels (*Robinson, 1984*; *Ganzhorn & Wright, 1994*; *Chapman & Chapman, 1991*). Given the size of Tioman,

and the logistical challenges of observing flying foxes foraging, the best way to overcome this possible information gap is to conduct GPS tracking studies.

We also acknowledge that NGS approaches to diet identification are semi-quantitative because chloroplast abundance is variable in different plant species and different parts of the leaf. Ultimately, the ability of NGS to accurately identify food plants will always depend on sequence specificity of the primers. While the NGS approach has proven to be useful in elucidating the island flying fox's varied diet on Tioman, for animals with such a diverse phytophagous diet, primer specificity will always be a limiting factor and there is a chance that unknown plant species will not be detected due to primer mispriming. Also, identical chloroplast DNA sequences can be present in different but related species, making it impossible to distinguish closely related plant species from each other in the diet. This could be one possible factor why several OTUs could only be identified to order/family/subfamily levels, suggesting that they require further phylogenetic investigation and/or may benefit from identification based on more rapidly-evolving plastid-coding genes such as *matK*. It is worth noting that the sequenced *rbcL* gene of some plant specimens collected in this study did not exhibit 100% identity matches to species in the BOLD/NCBI databases, which may be attributed to genetic diversity at the intra-species level or gaps in the database i.e., certain plant species consumed by the flying foxes may not yet have their corresponding sequences deposited in the database. Nevertheless, there is an urgent need for the BOLD/NCBI databases to have more representation of plant sequences from Peninsular Malaysia and Southeast Asia in general.

Another limitation of the NGS approach for generalist diets is that it does not identify which part of the plant was consumed. For specialised frugivores, nectarivores, or herbivores (e.g., folivorous), this may not be an issue. Flying foxes, however, are generalists which consume fruits, flowers, nectar, and even leaves (*Marshall, 1985*). It is this dietary plasticity which allows them to perform more than one ecological role in tropical landscapes. Therefore, identifying which plant parts are actually consumed is a crucial step towards identifying the ecosystem services that these bats provide. NGS can provide a first step towards identifying flying fox diet but should not be viewed as a replacement for microhistological analysis. Nevertheless, this approach has shed new light on flying fox diet by discovering plant taxa that were entirely missed out by the conventional approach. Ideally, studies using NGS should be combined with micro-histological analysis in order to fill in the gaps and broaden our understanding of pteropodid diet and foraging ecology. NGS can also be used in combination with comprehensive and long-term data on plant phenology, to observe which food resources are available at which time. Following on from this preliminary study, the identification of specific food plants via NGS can now help guide more in-depth plant sample collection and phenological observations.

## CONCLUSION

Our study is the first to use NGS to identify potential plant species in flying fox diet, paving the way for a new approach to studying flying fox diets. Since our NGS analysis of flying fox diet was semi-quantitative, it is not yet possible to make any definite conclusions regarding

food preference vs. food availability; ultimately it is unclear to what extent sampling bias and detection probability may have influenced the type and relative abundance of plant taxa detected in our study. Yet some of the interesting patterns we observed are worth investigating in greater detail, particularly in combination with microhistological analysis. The results will also help to guide us in conducting more accurate and expanded phenology monitoring, and further collection of botanical samples. Further and more rigorous sampling, especially at the level of the individual animal, is required to understand the dietary patterns of this particular flying fox population, expand on the information provided here and build on our understanding of how flying foxes provide ecosystem services on Tioman Island and elsewhere.

## ACKNOWLEDGEMENTS

We thank the people of Juara for their hospitality and support. We are grateful to the Economic Planning Unit of Malaysia (Permit No. 3242) and the Department of Wildlife and National Parks (DWNP) for allowing this research to be conducted. In particular, we thank Dr. Jeffrine Rovie Ryan Japning and Frankie Thomas Sitam for the use of DWNP lab facilities. Special thanks to Lam Wai Yee, Esteban Brenes-Mora, Anna Deasey, Noraisah Majri, Mahfuzatul Izyan, Khatijah Haji Hussin, Joanne Tong, Yek Sze Huei, Liz Moleski, Sri Rao Venkateswara, Foon Junn Kitt, Jackie Tan May Li, David Bickford, Mary Rose Posa, Lim Lee Sim, Jasdev Sohanpal and Manpreet Kaur for assisting with the collection of droppings. We are indebted to Sara Bumrungsri for training in dropping collection and microscope analysis, and Kartini Mohamed and Muhamad Hamirul for facilitating the use of light microscopes. We are also grateful to two anonymous reviewers and Johan Ponsu for providing constructive feedback to improve the manuscript.

### Funding

This work was supported by The Rufford Foundation (grant number 17325-1), Bat Conservation International (Scholarships and Grassroots Grants for Bat-Centric Projects Focused on Critical Conservation Needs), Muséum National d'Histoire Naturelle, University of Nottingham Malaysia Campus, Dr. Yeoh Suat Hui from Universiti Malaya, and Ms. Jade Ong. SA Aziz was the recipient of a PhD scholarship from the French Government issued by the French Embassy in Malaysia. GR Clements was supported by a LABEX BCDiv grant while in France. There was no additional external funding received for this study. The funders had no role in study design, data collection and analysis, decision to publish, or preparation of the manuscript.

### Grant Disclosures

The following grant information was disclosed by the authors:
The Rufford Foundation: 17325-1.
Bat Conservation International.

Muséum National d'Histoire Naturelle, University of Nottingham Malaysia Campus.
Universiti Malaya.
French Embassy in Malaysia.
LABEX BCDiv.

## Competing Interests

Sheema Abdul Aziz and Gopalasamy Reuben Clements are co-founders of Rimba, Selangor, Malaysia.

## Author Contributions

- Sheema Abdul Aziz and Han Ming Gan conceived and designed the experiments, performed the experiments, analyzed the data, contributed reagents/materials/analysis tools, wrote the paper, prepared figures and/or tables, reviewed drafts of the paper.
- Gopalasamy Reuben Clements conceived and designed the experiments, performed the experiments, wrote the paper, prepared figures and/or tables, reviewed drafts of the paper.
- Lee Yin Peng performed the experiments, analyzed the data, wrote the paper, prepared figures and/or tables.
- Ahimsa Campos-Arceiz, Kim R. McConkey and Pierre-Michel Forget reviewed drafts of the paper.

## Field Study Permissions

The following information was supplied relating to field study approvals (i.e., approving body and any reference numbers):

This research was approved by the Economic Planning Unit of Malaysia (Permit number: 3242).

## DNA Deposition

The following information was supplied regarding the deposition of DNA sequences:

Sanger sequencing results for the rbcL fragment of collected plant specimens were assigned accession numbers in GenBank/NCBI (see Table S1). NGS data were also deposited in the SRA database as mentioned in the Data Availability Section.

Accession numbers:

SRX1988609, SRX1988608, SRX1988607, SRX1988606, SRX1988605, SRX1988604, SRX1988603, SRX1988602, SRX1988601, SRX1988600, SRX1988599, SRX1988598, SRX1988597, SRX1988596, SRX1988595, SRX1988594, SRX1988593, SRX1988592, SRX1988591, SRX1988590, SRX1988589, SRX1988588, SRX1988587, SRX1988586, SRX1988585, SRX1988584, SRX1988583, SRX1988582, SRX1988581, SRX1988580, SRX1988579, SRX1988578, SRX1988577, SRX1988576, SRX1988575, SRX1988574, SRX1988573, SRX1988572, , SRX1988570, SRX1988569, SRX1988568, SRX1988567, SRX1988566, SRX1988565, SRX1988564, SRX1988563, SRX1988562, SRX1988561, SRX1988560

https://www.ncbi.nlm.nih.gov/sra/SRP080299.

## Data Availability

The research in this article only generated DNA sequences, which have already been listed in detail in the relevant section above. No other raw data or code were generated.

## Supplemental Information

Supplemental information for this article can be found online at http://dx.doi.org/10.7717/peerj.3176#supplemental-information.

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
