# Peer review of "Elucidating the diet of the island flying fox (Pteropus hypomelanus) in Peninsular Malaysia through Illumina Next-Generation Sequencing"

_PeerJ, doi:10.7717/peerj.3176_

## Round 0.1 · original submission · Major Revisions

We have received detailed reports from three reviewers on your article. All three reviewers and I see considerable merit in your study and significant general interest in the application of NGS to study the diet of endangered and elusive wildlife species. However, your paper would clearly benefit from careful revisions to address the fairly length list of editorial changes mostly pertaining to the provision of adequate detail and rationalization of the approaches taken in your study. While I don't see any particular major technical shortcomings, the list of minor ones is substantive and will require considerable attention.

Reviewer 1 ·

Basic reporting

The manuscript of Abdul Aziz and colleagues deals with an interesting subject and uses powerful methods to assess the diet of a threatened species.

Overall, the paper is clearly written although further proof reading by a native-speaker would help to make the phrasing more concise and precise. I found the manuscript too long in particular the description of the methods (ex : >40 lines for the description of the sampling strategy alone!) which contains too many redundancies and irrelevant details. The discussion also needs to be streamlined with more sub-sections.

Figures /Captions

There are too many figures and tables. Some (Figure 2, Figure 4, Figure 6, Table 1 and 2) would be best placed in suppl. material. Some tables/figures are not appropriately described or labeled : the caption of the Figure 3 is not enough informative and the axis labels are missing. For the sake of clarity, the taxonomic identification should be specified in this figure (i.e. name of the families identified). Perhaps, the authors should report only the proportions of reads from the most frequent OTU and gathered the less frequent ones (<10-15%) in a single class. The Figure 4 (phylogenetic tree) cannot be published in this state : Here also, the caption is incomplete and the taxon names and boostrap values are written in too small letters. I guess that the sequence ID marked in red correspond to the rbcl sequences obtained from collected leaf samples but this needs to be specified in the caption and the authors should have reported instead the name of the corresponding taxon.

Literature reference

Overall, literature is appropriately referenced. However, the methods and the parameters used for the NGS analysis sometimes lack of justification (L265-283) (see below).

L.102. Here, the authors might also cite Quéméré et al. Plos One (2013).

Experimental design

The paper seems analytically correct and I just have minor comments regarding (i) the absence of justification/comparison with other methods traditionally used to analyze such data and (ii) the criterion used to build clusters/clades of sequences that may lead to inaccurate taxonomic identification when the level of completeness of the reference database is weak for a family (see below).

(i) There is now a large body of studies assessing diet in herbivores using DNA metabarcoding and most used a different barcoding maker (trnL vs rbcl) and a different methodology to denoise the dataset and identify taxon (i.e. identify a unique sequence for each food plant and filter sequencing errors/chimeric sequences). I found interesting the authors attempt to use a phylogenetic approach to delineate and name OTU but they must make further effort to clarify the different steps of their methodology (see below) and discuss about the benefit/limitation compared to other approaches. Why did they chose this barcode ?
Does “rbcl” have a better discrimination power than the widely-used “trnL” barcoding marker for the plant families consumed by flying fox ?
Did the rbcl have a higher level of completness in the online reference databases ? How they dealt with the numerous sequencing errors expected using NGS ?

L113. “Infer spatio-temporal dietary patterns” This objective need to be clarified since only two locations were compared.

L.226. the authors need to specify if these primers (rbcLa) have been designed specifically for this study or otherwise cite a reference.

L.271. This is a big advantage compared to the non-coding trnL.

L276. Did the OTUs were also searched against Genbank/NBCI as stated in the table 3 ?

L278. “The top non-redundant 10 Blast hits… were used for phylogenetic construction”. But how the authors figured out the cases where there were more than 10 blast hits with identical match scores? Why did not they use a minimum identify threshold/match score ? My concern is that this arbitrary criterion may lead the accuracy of the taxonomic identification to be strongly related to the level of completeness of the reference database for a specific plant family/part of the phylogenetic tree. Some taxa may be wrongly identified when the closely related taxa within the reference database belong to a different family than the generated sequences. To identify such case, the authors must first investigate the discrimination power of trnL for each plant family consumed by flying foxes.
L311. “After the filtration”. This needs to be clarified.

L312. It’s not clear how the taxa (genera/families) were identified based on the phylogenetic tree. Did they use bootstrap support values ?

L347-350. This should be in the discussion section.

L363-365. This should be mentioned in the results. Furthermore, all the sequences produced for this study should be deposited in an online database (even if none match up with the plants parts found the flying fox droppings).

Validity of the findings

The authors properly discussed the various limitations of their study and are cautious in their conclusion given these limitations. Subsections are required to make the reading of the dicussion easier. The authors must provide the accession number for NGS data in the manuscript.

Reviewer 2 ·

Basic reporting

The manuscript entitled “Elucidating the diet and foraging ecology of the island flying
fox (Pteropus hypomelanus) in Peninsular Malaysia through Illumina Next-Generation Sequencing” describes diet analyses of island flying fox (Pteropus hypomelanus) in Malaysia and the effectiveness of NGS to this end. This manuscript is well written and structure of the manuscript conforms with PeerJ standards. With regard to raw data, the authors have submitted the rbcL plant sequences to GenBank and have provided the accession numbers, and have given raw data on individual OTUs. It would be good to submit the metabarcoding sequence data in a public repository.

Experimental design

Overall experimental design is able to address the question of interest: which is to assess utility of NGS to characterize diet, assess spacio-temporal patterns and compare NGS based diet analyses with morphological characterization. It is furthermore good to see more work on Southeast Asian mammal species. However, there are methodological issues that need to be addressed or caveats stated in the manuscript.

1) From my understanding the primer designed for the study was based on alignment of rbcL sequences generated from the reference database of preferred food plants or its related species (this is a bit unclear as the methods (L241) don’t state which alignment was used for primer design “Based on the alignment, primers targeting 220bp of rbcL gene were designed using Primer3”: this should be clarified). If this was the case, the new primers might bind preferentially to known diet items and not to unexpected ones. This may create two problems (1) bias identifications towards known species (2) underestimate the importance of unexpected species in the diet. Note that there are established primer pairs such as trnL g-h as mentioned or rbcL minibarcode (Little 2013 doi:10.1111/1755-0998.12194) that have been developed and tested with more universal datasets. Given that there is a new primer involved, and this bias has not been tested in the manuscript, the caveat and implications should be discussed in the discussion.

2) OTU search is usually done with a distance threshold, but this is not mentioned. This can be important in determining whether multiple species might be grouped. Plant barcodes are more conserved than animal barcodes and multiple species barcodes can often be identical or very similar. This will influence dietary diversity.

3) It should also be clarified which of the various OTU sequences was taken as the “representative sequence” that was used to build the Maximum Likelihood phylogeny (the most dominant sequence as the representative?).

4) For OTU to taxonomic identification Lines 276-79 state “remaining OTUs were searched against BOLD database and reference sequences obtained from this study to obtain identity of the OTUs. The top non-redundant 10 BLAST hits for each OTU along with the generated rbcL sequences from this study were used for phylogenetic reconstruction.” It is not mentioned how the sequences were searched against reference sequences obtained in the study. Also, Table 3 has hits to both BOLD and GenBank but only BOLD is stated in methods.

5) It should also be explained how Table 3 is made in the methods: It is surprising to me that closest match using a short fragment of rbcL are species specific, especially from GenBank. It is my understanding that several species share rbcL sequences. Is this the case in the present study or is it that the “top” hit is represented? Note that the “top hit” would not be a good summary if identical matches are found and reporting this only can be misleading. For this one could use a “Lowest Common Ancestor” approach, i.e., if identical best matches are found to members of same species/genus/family/… they are summarized at that taxonomic hierarchy. Automated tools such as MEGAN or readsidentifier are available for this and can be used if manual inspection is not possible.

Validity of the findings

No additional comments

Additional comments

Other minor comments:

Line 100: “Non-invasive DNA analyses of faeces”: Instead of referring to DNA analyses as non-invasive, it may be better to state that faecal samples are collected in a non-invasive manner

Line 104 “this has never before been attempted for pteropodids or plant-based mammal diets in the Paleotropics”: This is not the case, see Srivathsan et al. 2016 (doi:10.1186/s12983-016-0150-4), for diet analyses of a primate in Southeast Asia.

Line 160: “feasibility of extracting plant DNA”: Authors might consider editing to “feasibility of amplifying plant DNA from DNA extracted from Pteropus droppings”, as the DNA extraction is not selective to plants.

Line 217-8: Edit to “When the individual of a plant matching the list of genera was”

Line 227/Table 2: A few suggestions for Table 2: The legend states “Primers used in this study for the amplification of rbcL from flying fox droppings. Bold, target sequence; underlined, Illumina partial adapter”. However, the first two primer pairs rbcLaf-M13 and rbcLa-revM13 are used for plant DNA barcoding and not droppings I believe? Secondly the M13 part of sequence is underlined, is this an Illumina adapter sequence? Kindly cite the rbcLaf and rbcLa-rev primers.

Line 379: It is stated that partial rbcL is highly accurate for family level identifications. This should be cited.

Line 389-390: Or additionally there is the possibility of using metagenomic shotgun sequencing to get whole genomic DNA characterized from select fecal samples and get multiple genes sequenced (Srivathsan et al. 2015, 2016).

Line 395: Other possibility for lack of 100% identity is some intraspecific variation.

Figure 2: Is the target length 198 bp or 220 bp as stated in line 241?
Figure 4: It makes it easier for the reader if it is clarified in the legend that PTMN IDs refer to barcodes generated in the present study.

·

Basic reporting

This study presents an application of an NGS approach to determine the diet of the island flying fox in an island of the Malay peninsula. Although more and more applied, this study is, to my knowledge, the first to apply an NGS approach (otherwise called "metabarcoding") to identify the diet of phytophageous bats. The discussion highlights several interesting points about limits of the approach described in the manuscript and challenges for future applications. However, there are some issues (listed below) that should be, at least, clarified before publication, especially about the sampling design and data analyses.

Experimental design

Line 191: Please describe the approach used in the field and in the lab to avoid environmental contamination as well as cross-contamination.
Lines 194-196: Why a such sorting of droppings? How many dropping per roost and per day did you pool?
Lines 199-203: Please precise if it is a homemade protocol for DNA preservation and storage. If not, add references.
Lines 226-227 and 247-257: Add references for rbcL primers that have been developed previously. A new primers pair has been designed in this study. How its discriminative power and its taxonomic coverage have been evaluated? Also describe in the main text the difference between primers pairs used. In the current form it is quite confusing (IlluM_ rbcLF is the same primer as rbcL-357F with a sequencing adapter but what is rbcLaf-M13?).
Lines 236: Why did you pool droppings per roost? By this way, you lose the individual dimension of the diet which might be interesting to characterize the foraging behavior.
Lines 239-241: it exists DNA kits specifically designed to extract DNA from feces. Using a plant gDNA extraction kit might be not adapted.
Line 297: Histological reference collection contains pollens of multiple species in the area but only plant parts for fig. This may induce a bias towards the detection of fig and limit the identification of other plant species.
Lines 316-320: It would be interesting to know how many droppings have been pooled each month. Trends in richness and even relative abundance can be influenced by a high difference in sampling effort.

Validity of the findings

Overall, results are mostly qualitative, some (basic) statistics would be welcomed.
Line 323: As sequencing depth may vary between samples, it would make more sense to talk in relative read abundance per sample rather than in an absolute number of reads. Also please provide some statistic about sequencing (number of reads and OTUs before/after filtering, mean number per sample etc.).
Line 310-311: This sentence is quite speculative at this stage because rare sequences can be degraded and not recovered.
Line 360: The discriminatory potential of this primer and its taxonomic coverage have not been formerly tested. The second parameter is highly important because some taxa might not be amplified if primers are not sufficiently conserved among plants. Some bioinformatic tools exist for this (e.g Riaz et al. 2011).
Line 325: 100 reads overall or per sample?
Line 349-351: It would be interesting to use replicates in this case to disentangle the part of variance that is due to biological variation and those coming from technical biases.
Supplementary material: The number of reads per sample (as shown in SI) seems very low for NGS data. It looks to be rarefied date but the total of each sample is 100 reads while according to the main text, the dataset is supposed to be normalized to 10000 reads? A "readme" should be added to help the reader to understand this table.

Additional comments

Abstract: To clarify the link between diet and ecosystem services, you should give examples of ecosystems services provided by bats.
Introduction: The introduction looks relatively disjointed. I would suggest a reorganization, for example talking about limits of traditional approaches (lines 55-72) just before the paragraph about molecular approaches (line 99).
Lines 65: Define "ejecta" here and maybe show an example on Fig.1. Also, this sentence is unclear: fecal collection does not require direct observations nor GPS collars. Please reformulate.
Line 79-82: Are they known to be an important seed disperser at the island level or between islands?
Line 134-135: To be consistent, please describe the population size in Tekek.
Line 137: Place the "study site" section before the "study species"
Line 282: I guess "(-nt -gtr)" are parameters, please describe what does that mean.
Line 311: "filtering" instead of "filtration"
Line 314: A new record compared to histological data or literature? Please add a reference
Line 314-315: Since authors made a taxonomic assignment, I do not really see the interest of making a phylogenetic tree based on "best hits" (and more generally of having the Fig. 4 because it is not discussed).
Line 327: What do you mean by "apparent"?
Line 361: A reference about the "known" diet of pteropodid is necessary here.
Line 401-403: Authors can also here problems related to PCR amplification biases, potential differential digestibility etc.
Figure 3: Please indicate the taxonomic assignment of MOTU, it will give more interest to this figure
Table 3 and Fig. 6 can be placed in supplementary material.

---

## Round 0.2 · Minor Revisions

Two of the original reviewers have assessed your revised manuscript and I am pleased to see that there is a consensus that your paper has been improved greatly through revision. As you will see from their comments there are a few more areas that merit further clarification and these are mostly minor (exc. comment from Reviewer 1 regarding the minimum homology threshold).

Reviewer 1 ·

Basic reporting

The authors have greatly improved the clarity and coherence of the ms. The objectives and methods are more clearly exposed and the authors made a real effort in terms of concision. The ms is now broadly well written but I have a list of minor comments which remain to be addressed before publication :
Lines 129-131 and lines 133-134. Please put these sentences in the next section regarding 'the study species'.
L135-136. Please remove this sentence (not informative if you don't cite the species names).
L144. rewrite the sentence as following « however populations are decreasing rapidly and the species is now listed as endangered on the Malaysian Red list (DWNP 2010) »
L145-148. I would remove this information that is useless for the understanding of the study.
L151. Repetitive with lines 133-134
L94-200. Analysis overview. I would have put this paragraph at the end of introduction (to briefy introduce the methodology employed)
L236 very important : please specify the sequences of the two primers designed.
L246-281. For the sake of clarity, please split this section in two parts entitled « laboratory procedure » and « sequence analysis and taxon assignation ».
L276. Please specify which homology threshold was used to select sequences imported in MEGAN ? (see my majorn comment below). Furthermore, you need to specify how the LCA method works (at least the general principle) and explain better how you assign taxon names.
L282. Please change the title to « Microhistological analysis » since there is nothing about the comparison between the two approachs in this part.
L311-312. Remove this sentence (not enough informative)
L320-321. I am bit surprised by the lack of taxonomic resolution of the marker (compared to my experience with the trnL). Please confirm that the LCA method has been applied solely on BOLD sequences that perfectly matched (homology 100%) with sequences in feces or that matched with a high score (for example >0.98) ? The homology threshold is not specified so I have some concerns : please confirm that you did not use the full list of sequences from for example the first page of outputs of blastn (with various matching scores).
Line 362. Replace « feasible » by « reliable » in the title.
L376. Please qualify this statement : the trnL is indeed shorter but it has probably a higher discrimination power than the marker used here probably because it is located in a non-coding region of the gene and evolved faster than rbcl (in most plant families).
Here, the authors should insist further on two main advantages of this barcode : its phylogenetic signal and the large/growing online database.
L383. Little 2014 (instead of 2013)
L387. Please note here that some sofware such as « ecoprimers » (see pipeline obitools) can be used to perform in-sicilo PCR from sequences in online database to test the reliability/universality of the new designed primers.
L391. « combination of target regions » Please clarify this statement.
L427-429. The syntax is approximative and I’m not sure to understand this idea. Please clarify.

Experimental design

Methods are now clearly described except the criterious used to assign taxon name to sequences in feces.

Major comment : It remains unclear which minimum homology threshold has been used (100%, 98%, 95%?) to select the sequences (from Blastn outputs) which are then analyzed in MEGAN. it's a criticial point because this may potentially affect the discrimination power of the method. I am surprised that so many sequences are identified at the family level only. Is it due to a the use of a too low homology threshold or to the slow evolution of the marker with these families ? Overall, the authors should clarify the procedure employed to assign taxon name.

Validity of the findings

The authors introduce a novel approach (alternative to the widely-used trnL approach) and discuss its advantage and limitations. The results are well supported and the discussion is cautious with appropriate references to literature and a part regarding potential flaws and perspectives.

Reviewer 2 ·

Basic reporting

The revised manuscript has addressed the various issues raised previously and reads well. I have a few minor suggestions listed in the general comments section

Experimental design

NA

Validity of the findings

NA

Additional comments

1) L106-L109: The two sentences seem contradictory, could the authors clarify? If the Hayward 2013 study describes molecular analyses for diet of a fruitbat, one cannot state that molecular analyses has been done only for insectivorous bat species.
2) L110: Would be more precise to restructure as "We evaluated the utility of Illumina Next Generation Sequencing (NGS) to identify plant species present in the droppings of the island flying fox from Tioman Island in Peninuslar Malaysia"
3) L230: It isn't clear how "high" the similarity is and what was the criteria for mining
4) L250 vs L246 and throughout text, kindly keep the spacing between the number and unit consistent.
5) L276-L278: Kindly specify the parameters for BLAST (blastn/megablast/.., evalue). There should be some percentage threshold used to exclude any sequences that didn't match well to the database (if 97% was the OTU delimitation criteria, similarly, 97 % identity criterion can be used) and low percentage hits should be removed before applying LCA. Authors may consider putting the % identity value in Table 1.
6) L304-6: I think a major factor is that it is based on a single genetic marker
7) L321: Edit to: 8 OTUS matched with specimens
8) L374-6: The argument that rbcL has a good database is valid as it is among the standard plant DNA barcode, but the argument that rbcL is an alternative to trnL P6 loop due to low taxonomic resolution needs more support. If no data is available for this, this statement should be excluded. The P6 loop of trnL, even though short is highly variable. While it may not be very good at taxonomic resolution, I am not sure how it compares with the 198 bp fragment of rbcL used here.
9) Figure 2 heading: Edit to Relative abundance of OTUs detected in flying fox droppings: From my understanding the graph uses data from all OTUs, but condenses several as one category. Hence I think the summary is not based on 8 OTUs only.
10) Can the title of Supplementary Figure S4 be bit more descriptive?

---

## Round 0.3 · accepted · Accept

The revisions appear to fully address the last round of reviewer comments. Thank you for your diligence and we look forward to seeing your paper in PeerJ.